# GENERATE TO DISCRIMINATE:
# EXPERT ROUTING FOR CONTINUAL LEARNING

## ABSTRACT

In many real-world settings, norms, regulations, or economic incentives permit the sharing of models but not data across environments. Prominent examples arise in healthcare due to regulatory concerns. In this scenario, the practitioner wishes to adapt the model to each new environment but faces the danger of losing performance on previous environments due to the well-known problem of *catastrophic forgetting*. In this paper, we propose Generate-to-Discriminate (G2D), a novel approach that leverages recent advancements in generative models to alleviate the catastrophic forgetting problem in continual learning. Unlike previous approaches based on generative models that primarily use synthetic data for training the label classifier, we use synthetic data to train a domain discriminator. Our method involves the following steps: For each domain, (i) fine-tune the classifier and adapt a generative model to the current domain data; (ii) train a domain discriminator to distinguish synthetic samples from past versus current domain data; and (iii) during inference, route samples to the respective classifier. We compare G2D to an alternative approach, where we simply replay the generated synthetic data, and, surprisingly, we find that training a domain discriminator is more effective than augmenting the training data with the same synthetic samples. We consistently outperform previous state-of-the-art domain-incremental learning algorithms by up to 7.6 and 6.2 points across three standard domain incremental learning benchmarks in the vision and language modalities, respectively, and 10.0 points on a challenging real-world dermatology medical imaging task.

## 1 INTRODUCTION

In continual learning, we would like to adapt our model to each new environment (*forward transfer*) in a sequence of tasks, while retaining the ability to predict accurately on data drawn from previous environments (*backward transfer*) (Parisi et al., 2019). However, naive sequential training on each new dataset can result in failures of backward transfer (sometimes referred to as *catastrophic forgetting* (McCloskey & Cohen, 1989; French, 1999)), where adaptation to new tasks coincides with performance degradation on previously seen environments.

We study the domain-incremental setting, where the set of possible labels is the same across environments and the goal, after each adaptation, is to produce a system that performs well at test time on examples drawn at random among all previously seen domains. Crucially, domain identifiers are not given at test time. But for the prohibition on data sharing, one could simply perform experience replay, training on the union of all available data (Chaudhry et al., 2019b). However, these approaches are often not viable in many real-world scenarios, where data regulations and security requirements create formidable obstacles to sharing data across institutions. In particular, we are motivated by healthcare applications such as medical imaging and risk prediction, where performance drops have been noted across institutions and time periods (Zech et al., 2018; Pooch et al., 2020; Guan & Liu, 2021; Otles et al., 2021; Finlayson et al., 2021; Zhou et al., 2023). These performance drops owe to diverse causes, including differences in patient demographics, scanning equipment, and image acquisition techniques. Moreover, these medical settings exhibit the key aspects of the continual learning setup: model sharing is often permitted, even in settings where data sharing is not.

Current approaches to this problem involve either training a single comprehensive classifier model with constraints to help mitigate forgetting (e.g., EWC Kirkpatrick et al. (2017)) or training distinct

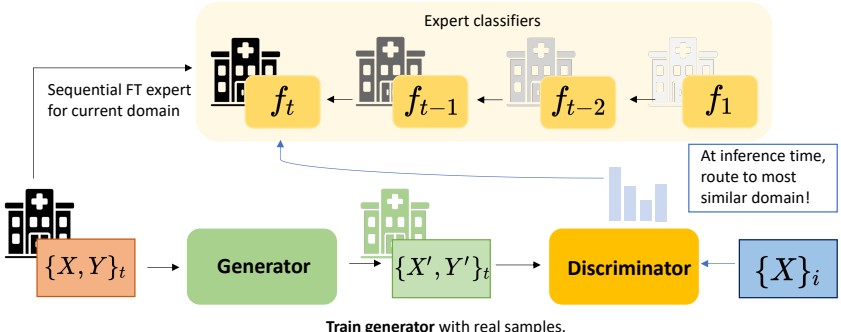

Figure 1: Generate to Discriminate (G2D); At train time, we i) fine-tune the generator and expert classifier and ii) train a domain discriminator on synthetic images produced by our generator. At inference time, based on our discriminator's prediction, we route test samples to the corresponding expert.

domain-specific classifiers, referred to as *experts* (as in Aljundi et al. (2017)). Recent advances in parameter-efficient fine-tuning techniques have enabled the training of domain-specific experts, such as prompts (Wang et al., 2022a), without significant additional storage requirements, surpassing previous domain-incremental learning methods. However, these methods necessitate an inference-time routing mechanism for expert selection and often rely on the implicit domain discriminative capabilities of pre-trained models (Aharoni & Goldberg, 2020). Nevertheless, this simplistic approach may not consistently perform well across all datasets (see §5.1). Prior works have also explored generative methods to create synthetic examples for experience replay (Shin et al., 2017; Sun et al., 2020; Qin & Joty, 2022). However, these approaches have largely under-performed state-of-the-art discriminative approaches, (see Sun et al. (2020), results in §5.2), due at least in part to the introduction of noise in the form of low quality synthetic examples. An intriguing question arises: Rather than employing noisy synthetic examples for generative replay, can they serve a more effective purpose in domain discrimination? More specifically, can we leverage synthetically generated samples to develop an inference-time routing mechanism?

In this work, we propose Generate-to-Discriminate (G2D), a continual learning method that leverages modern generative models — conditional diffusion models or language models — to generate per-domain synthetic examples for purposes of domain discrimination (rather than generative replay). We then leverage this discriminator to route each example to the best expert. Concretely, for each new domain, we: (i) fine-tune an expert classifier; (ii) fine-tune a generative model and sample synthetic examples for when we no longer have access to real data; and (iii) train a domain discriminator to predict which domain a given sample is drawn from using generated samples from previous and current domains. At inference time, we pass samples through our domain discriminator, which routes each sample to its corresponding expert classifier.

Our experiments demonstrate that G2D outperforms previous state-of-the-art replay-based, regularization-based, and rehearsal-free prompt-based methods by up to 7.6 and 6.2 points on existing benchmarks across vision and language modalities, respectively. Interestingly, we find that competitive continual learning baselines, such as recent prompt-based methods, underperform in some real-world settings (see results in §5.1), emphasizing the need for more diverse and realistic benchmarks. Towards this end, we further introduce a new publicly available benchmark consisting of a sequence of four dermatology medical imaging classification tasks. Our approach outperforms previous work by 10.0 points on this pragmatic new challenge set.

In summary, we contribute the following:

- Generate-to-Discriminate (G2D), an expert routing method for domain-incremental learning using generative models for domain discrimination.
- Analysis demonstrating that training a domain identifier outperforms augmenting training data for downstream classification with the same synthetic samples (i.e., the generative replay approach).

- A new continual learning benchmark in the medical domain consisting of a sequence of four dermatology medical imaging classification tasks.
- Experiments demonstrating that our method outperforms previous domain-incremental learning approaches up to 7.6 and 6.2 points on standard benchmarks for both vision and language modalities and 10.0 points on our new medical imaging task.

## 2 RELATED WORK

### 2.1 CONTINUAL LEARNING AND DOMAIN-INCREMENTAL LEARNING

We focus on the domain-incremental setting of continual learning (Van de Ven & Tolias, 2019). In this setting, the learner must adapt to new domains while performing well on previously seen domains. Most continual learning methods fall into the categories of (i) parameter-based and (ii) data-based regularization techniques (Sodhani et al., 2022). Parameter-based regularization techniques – Two notable methods within this category are Elastic Weight Consolidation (EWC; Kirkpatrick et al. (2017)) and Synaptic Intelligence (SI; Zenke et al. (2017)). Both EWC and SI assess the importance of parameters related to previous domains and utilize a penalty term to safeguard the knowledge stored in those parameters while updating them for new domains. Learning without forgetting (LwF; Li & Hoiem (2017)) is another parameter-based regularization method, where knowledge of previous tasks are preserved by using the initial task knowledge as a regularizer during training.

Data-based regularization techniques – these approaches retain a subset of data from previous domains as an episodic memory, which is sparsely replayed during the learning of new domains. Several replay-based methods have been proposed, each differing in whether the episodic memory is utilized during training, such as GEM (Lopez-Paz & Ranzato, 2017), A-GEM (Chaudhry et al., 2019a), ER (Chaudhry et al., 2019b), MEGA (Guo et al., 2020), or during inference, like MbPA (de Masson D'Autume et al., 2019; Wang et al., 2020). These methods assume that the true data can be retained for replay. However, they cannot be used in settings where data sharing is restricted. To address this limitation, deep generative replay-based methods have been introduced (DGR; Shin et al. (2017), LAMOL; Sun et al. (2020), LFPT5; Qin & Joty (2021)). The main idea behind these methods is to learn a generative model of the data and use it to generate samples for experience replay. Additionally, there have been recent works investigating conditional generative replay methods, using GAN-based and VAE architectures Van de Ven et al. (2020); Zhao et al. (2022); Lesort et al. (2019).

In response to the increasing popularity of pre-trained models, another approach has emerged in the field of continual learning. Mehta et al. (2023) demonstrate that pre-trained initializations implicitly mitigate the issue of forgetting when sequentially fine-tuning models. Another line of approaches, known as prompt-based continual learning, exemplified by L2P (Wang et al., 2022c), DualPrompt (Wang et al., 2022b), S-Prompt (Wang et al., 2022a), and CODA-Prompt (Smith et al., 2023), involves learning a small number of parameters per domain in the form of continuous token embeddings or prompts while keeping the remaining pre-trained model fixed. The appropriate prompt is then selected based on the input data. Although these methods allow for continual learning without rehearsal, they depend on access to pre-trained models that provide a high-quality backbone across all domains, which may not be available in sensitive environments in real-world deployment (e.g., healthcare). Another classical approach involves incorporating task-specific experts for each new task in a sequence and subsequently using an expert gate to direct examples to the appropriate expert (Aljundi et al., 2017). Like prompt-based methods, this approach relies on the inherent domain discriminative abilities of AlexNet (pre-trained with ImageNet) and is susceptible to the same limitations mentioned earlier in the context of prompt-based methods.

### 2.2 VISION AND LANGUAGE GENERATIVE MODELS

We fine-tune text-to-image Stable Diffusion (Rombach et al., 2022) for vision modality and T5 for language modality (Raffel et al., 2020). Conditional image generation has been a prominent research area, mainly concerning contributions from Generative Adversarial Networks (GANs; Goodfellow et al. (2014)), Variational Autoencoders (VAE; Kingma & Welling (2022)), and more recently, diffusion models (Ho et al., 2020; Dhariwal & Nichol, 2021). Stable Diffusion is a text-to-image latent

diffusion model (LDM; Rombach et al. (2021)) conditioned on text embeddings of the class label. A diffusion model first learns a forward pass by iteratively introducing noise to the initial image. Subsequently, a backward pass eliminates the noise, recovering the final, generated image. To incorporate text conditions, a cross-attention mechanism is integrated into the U-Net architecture. This allows the infusion of textual information during the image generation process. T5 (Raffel et al., 2020) is a text-to-text transformer model that addresses a wide variety of language tasks (e.g., translation, summarization, question answering) in one unified framework. T5 has been trained on C4, which consists of a cleaned version of Common Crawl. The T5 model has been demonstrated to have strong performance after finetuning on a wide variety of downstream tasks.

## 3 GENERATE TO DISCRIMINATE (G2D)

### 3.1 PROBLEM FORMULATION: DOMAIN-INCREMENTAL LEARNING

We focus on the *domain-incremental* continual learning setting, where the primary goal is to learn a model that adapts to each new domain while mitigating catastrophic forgetting on previously seen domains. Formally, we consider a sequence of $T$ domains, $\mathcal{D}_1 \rightarrow \cdots \rightarrow \mathcal{D}_T$, where $\mathcal{D}_t = \{x_i^t, y_i^t\}_{i=0}^{N_t}$ represents a dataset corresponding to domain $t$, sampled from an underlying distribution $P_t(\mathcal{X}, \mathcal{Y})$. $x_i^t \in \mathcal{X}$ is the $i$-th image or text passage and $y_i^t \in \mathcal{Y}$ is its label. $N_t$ is the total number of samples for domain $t$.

Furthermore, in the domain incremental scenario, $\forall t$, the marginal or conditional distributions over $\mathcal{X}$ and $\mathcal{Y}$ can change, i.e., $P_t(\mathcal{X}) \neq P_{t+1}(\mathcal{X})$ and $P_t(\mathcal{Y}|\mathcal{X}) \neq P_{t+1}(\mathcal{Y}|\mathcal{X})$, while the label space $\mathcal{Y}$ remains fixed across all domains. The goal is to learn a predictor $f_\theta : \mathcal{X} \rightarrow \mathcal{Y}$, parameterized by $\theta \in R^P$, to minimize the average expected risk across all $N$ domains. To demonstrate the model's learning behavior over the sequence of domains and analyze catastrophic forgetting of the previously seen domains, we evaluate the model after training on a specific domain $t$ using the test dataset of that domain, $\mathcal{D}_t^{test} \sim P_t(\mathcal{X}, \mathcal{Y})$, and from past domains, $\mathcal{D}_i^{test} \sim P_i(\mathcal{X}, \mathcal{Y}), \forall i \in [1, \ldots, t-1]$. During sequential training, the domain identity is known. However, during inference, the domain identity is unknown. Let $\alpha_{s,t}$ denote the accuracy on domain $s$ after training on domain $t$. Following prior work (Lopez-Paz & Ranzato, 2017), we compute the *average accuracy* ($A_t$) metric after training on the domain $t$. Formally, $A_t$ is given by $A_t = \frac{1}{t} \sum_{s=1}^t \alpha_{s,t}$.

### 3.2 GENERATE TO DISCRIMINATE (G2D)

**Generation of synthetic samples for domain discriminator.** At each domain $\mathcal{D}_t$, we fine-tune a generative model $G$ with samples from the current domain $\{x_i^t, y_i^t\}_{i=0}^{N_t}$. To perform finetuning, we use parameter efficient techniques, i.e., LoRA (Hu et al., 2021), where the only trainable parameters are low-rank matrices that are added to the attention layers, and for the text domain we use prompt tuning (Lester et al., 2021) which learns continuous input token embeddings. Then, we generate synthetic samples $\mathcal{M}_t$ from $G$. Given synthetically generated data from domains $\mathcal{D}_1, ..., \mathcal{D}_t$, we train a domain discriminator[1] $D_{\theta_t}$ on the union of the synthetically generated samples $\mathcal{M}_1 \cup \mathcal{M}_{t-1} \cup \mathcal{M}_t$, for domain identity prediction (i.e., $t$-way classification). More formally, we construct a dataset of $\bigcup_{i=1}^t \{(x, i)|x \in \mathcal{M}_i\}$. In essence, our domain discriminator learns to predict domain membership, or route samples to their corresponding or most similar domains.

**Expert classification models.** At each domain $\mathcal{D}_t$, we sequentially fine-tune our classifier $f_{\theta_t}$ as the expert on domain $t$, and add $f_{\theta_t}$ to our list of experts $\{f_{\theta_1}, ...., f_{\theta_{t-1}}, f_{\theta_t}\}$. At inference time, we use our domain discriminator to predict the most likely domain, and the test sample is routed to the corresponding expert classifier for our class prediction (see Algorithm 1).

## 4 EXPERIMENT SETUP

### 4.1 TASKS, DATASET, AND METRICS

In our vision experiments, we assess the effectiveness of our approach using the following datasets: DomainNet (Peng et al., 2019), CORe50 (Lomonaco & Maltoni, 2017), and DermCL, a newly in-

---

[1] In Appendix B, we include an ablation where the domain discriminator is also trained continually.

---

**Algorithm 1** Generate to Discriminate (G2D)

---

1: **procedure** ROUTING_ALGORITHM($D_T$, $f = \{f_{\theta_1}, \ldots f_{\theta_T}\}$, $x$)
2:      $domainIdx \leftarrow D_T(x)$         ▷ Predict the domain index for the given data point
3:      $f_\theta \leftarrow f[domainIdx]$                 ▷ Index into the list of expert models
4:      **return** $f_\theta(x)$                ▷ Run inference using the selected expert model
5: **end procedure**

---

troduced benchmark that we have curated from real-world dermatology tasks (Tschandl et al., 2018; Cassidy et al., 2022; Pacheco et al., 2020; Daneshjou et al., 2022). DomainNet is a domain adaptation benchmark, consisting of six distinct domains: real (photos), quickdraw, painting, sketch, infograph, and clipart (in order). Each domain has a fixed set of 345 classes and roughly 600,000 images. CORe50 is a continuous object recognition benchmark, consisting of multiple views of the same objects taken in different sessions (variations in background, lighting, pose, occlusions, etc). There is a sequence of 11 domains, where 3 domains are fixed as an out-of-distribution (OOD) test set for consistency, and the remaining 8 domains are used for sequential training. Our introduced DermCL benchmark consists of four domains of dermoscopic image datasets – HAM10000 (Tschandl et al., 2018), BCN2000 (Cassidy et al., 2022), PAD-UEFS-20 (Pacheco et al., 2020), and DDI (Daneshjou et al., 2022), for a multi-class classification task, over 5 unified labels of skin lesions. Distribution shifts between domains in datasets exist in terms of patient demographics, dataset collection period, camera types, and image quality. Following previous practice (e.g., DomainNet), we keep the task sequence fixed for consistency to HAM10000 $\rightarrow$ BCN2000 $\rightarrow$ PAD-UEFS-20 $\rightarrow$ DDI. For the above datasets, we report the average accuracy averaged over 5 random seeds. For DermCL, due to a high imbalance in label distribution, we report the average ROC AUC instead of the average accuracy.

In our text experiments, we evaluate our method on the standard domain-incremental question-answering benchmark as introduced by de Masson D'Autume et al. (2019). The benchmark consists of three question-answering datasets: SQuAD v1.1Rajpurkar et al. (2016), TriviaQA (Joshi et al., 2017) and QuAC Choi et al. (2018). TriviaQA has two sections, Web and Wikipedia, which are considered separate datasets. Following de Masson D'Autume et al. (2019), we process our dataset to include 60,000-90,000 training and 7,000-10,000 validation examples per domain. We use four different orderings of domain sequences (see Appendix A). Following prior works (de Masson D'Autume et al., 2019; Wang et al., 2020), we compute $F_1$ score for question answering task and evaluate the model at the end of all domains, i.e., we compute $A_4$.

### 4.2 BASELINE METHODS

We compare our method with state-of-the-art continual learning methods that address the domain incremental setting. First, we compare with elastic weight consolidation **EWC** (Kirkpatrick et al., 2017), a traditional, parameter-based regularization method. EWC constrains parameters to lie in regions of low error for previous domains by applying a penalty term determined by the Fisher information matrix.

In the context of data-based regularization methods, we compare to generative replay (**Generative Replay**), where a buffer of synthetic samples is used to train the label classifier. For the image domain, we implement our variant of generative replay using Stable Diffusion. We also compare with experience replay **ER** with limited examples (Chaudhry et al., 2019b), which maintains a subset of *actual* samples from previously seen domains in its buffer (which are not available in reality under data-sharing constraints).

For the vision datasets, we further compare with recent advances in prompt-based methods which have greatly boosted state-of-the-art performance across many benchmarks. The objective of such methods is to optimize prompts (i.e., small learnable parameters) to instruct the model prediction and explicitly manage task-invariant and task-specific knowledge. We compare our method with Learning to Prompt (**L2P**) (Wang et al., 2022c) and **S-Prompts** (Wang et al., 2022a), since they address the domain-incremental setting. L2P learns to dynamically prompt a pre-trained model to learn tasks sequentially under different task transitions. S-Prompts learns independent prompts per domain and employs a KNN domain identifier to route samples to the corresponding domain at

inference time and invoke the corresponding set of prompts (i.e., domain-specific model parameters). We note that there exist two variants of S-Prompts (ViT-based, and CLIP-based). For a fair comparison, we compare with the ViT-based (pre-trained ImageNet checkpoint) variant.

For our text experiments, we consider prominent baselines from prior work. Initially, we assess vanilla sequential fine-tuning (**SeqFT**), which does not employ any lifelong learning regularization techniques. We follow the precedent set in the work of de Masson D'Autume et al. (2019) for ER and Generative Replay, either retaining or sampling text examples in proportion to the dataset sizes. Additionally, we compare our approach with methods that leverage replay buffers for task-specific test-time adaptation, namely **MbPA++** (de Masson D'Autume et al., 2019) and **Meta-MbPA** (Wang et al., 2020). Meta-MbPA trains the model to attain a more suitable initialization for test-time adaptation and currently represents the state-of-the-art performance on the question-answering benchmark.

Lastly, we also compare against the setting of when access to real data from all domains is allowed at every task step, termed the multi-task learning (**MTL**) baseline. This is equivalent to training on the union of all existing data and can be viewed as an upper bound on performance when there is no significant negative transfer between domains.

### 4.3 Implementation details

**Vision.** For the vision domain, we fine-tune an off-the-shelf, text-to-image Stable Diffusion (Rombach et al., 2022) model, initialized with the pre-trained checkpoint. For parameter efficiency, we fine-tune our generator with low-rank adaptation (i.e., LoRA) (Hu et al., 2021)[2] where we only adapt the attention weights and keep the remaining parameters of the UNet architecture frozen. The architecture of our domain discriminator is a vision transformer (ViT B-16) (Dosovitskiy et al., 2020), initialized with the pre-trained ImageNet (Deng et al., 2009) checkpoint. The expert classifiers of our approach also follow the same architecture as the domain discriminator. See Appendix C.1 for additional implementation details.

**Text.** For our text generator, we use prompt tuning (Lester et al., 2021) to learn parameter-efficient models. We use the pre-trained T5-Large v1.1 checkpoint adapted for prompt tuning as the backbone (Raffel et al., 2020) and the prompt embeddings are initialized randomly. For training our domain discriminator and expert, we use low-rank adaptation (LoRA; Hu et al. (2021)) and freeze the pre-trained BERT-Base (Kenton & Toutanova, 2019) backbone. See Appendix C.2 for additional implementation and training details.

## 5 Experiments

### 5.1 Main results: comparison to other methods

Our method, G2D, consistently outperforms various domain-incremental learning baselines in all three vision benchmarks, as detailed in Tables 1, 2, and 3. Specifically, we observe significant absolute improvements of approximately 7.6, 6.0, and 10.0 points compared to previous state-of-the-art methods on DomainNet, CORe50, and DermCL, respectively.

We also note that our method demonstrates better out-of-distribution (OOD) performance compared to previous baselines, as evidenced by the results on CORe50 in Table 2, where evaluations encompass three OOD datasets. Table 3 illustrates our substantial advantage on DermCL compared to standard benchmarks like DomainNet and CORe50, where we outperform exemplar-free methods like L2P and S-Prompts. This underscores the need for more realistic benchmarks in domain-incremental learning. Prompt-based methods, while parameter-efficient, assume alignment between pre-trained models and downstream tasks, which may not hold in domains like medical imaging. This highlights the necessity for more comprehensive evaluations of methods in real-world scenarios. *Thus, we introduce DermCL to facilitate research in real-world evaluations and demonstrate the performance of our approach.*

For the text domain, our results on the question-answering benchmark are presented in Table 4. It is evident that our approach, with full fine-tuning, G2D Full FT (66.6), outperforms both ER (61.2)

---

[2]See Appendix C.5 for details on LoRA finetuning vs. full finetuning for the generator.

| Method | Average Accuracy (↑) |
|---|---|
| ER (50/class) | $52.79 \pm 0.03$ |
| EWC | $47.62^{\dagger}$ |
| L2P | $40.15^{\dagger}$ |
| S-Prompts | $50.62^{\dagger}$ |
| Supervised Contrastive - CaSSLe | $50.90^{\dagger}$ |
| Generative Replay (Ours) | $52.97 \pm 0.07$ |
| **G2D** | $58.45 \pm 0.56$ |
| **G2D (Full FT)** | $54.37 \pm 0.90$ |
| Upper Bound (MTL) | $64.36 \pm 0.04$ |

Table 1: Results on DomainNet. We compare performance in terms of average accuracy after training on the last domain (averaged over 5 random seeds). ↑ indicates higher is better and † denotes results obtained from Wang et al. (2022a). G2D (Full FT) is an ablation of our method, where we use vanilla full fine-tuning instead of the LoRA fine-tuning approach for all models. Highest performance is highlighted in green.

| Method | Average Accuracy (↑) |
|---|---|
| ER (50/class) | $80.10 \pm 0.56^{\dagger}$ |
| EWC | $74.82 \pm 0.60^{\dagger}$ |
| L2P | $78.33 \pm 0.06^{\dagger}$ |
| S-Prompts | $83.13 \pm 0.51^{\dagger}$ |
| Supervised Contrastive - CaSSLe | $75.68 \pm 0.60$ |
| Generative Replay (Ours) | $86.28 \pm 0.55$ |
| **G2D** | $89.11 \pm 0.30$ |
| **G2D (Full FT)** | $86.60 \pm 0.28$ |
| Upper Bound (MTL) | $94.56 \pm 0.12$ |

Table 2: Results on CORe50. We compare performance in terms of average accuracy after training on the last domain (averaged over 5 random seeds). ↑ indicates higher is better and † denotes results obtained from Wang et al. (2022a).

and test-time adaptation techniques such as MbPA++ (61.9) and Meta-MbPA (64.9) and G2D with parameter-efficient LoRA (64.7) is competitive with Meta-MbPA. It is important to note that all of these baseline methods retain actual samples in their buffers (1% of total samples). Similarly, when evaluating our approach on CORe50, it consistently outperforms methods that preserve real examples from previous domains, specifically ER and L2P with a buffer size of 50 samples per class (see Table 2). *These findings underscore the ability of our method to enhance performance even in scenarios characterized by stringent constraints on data sharing.*

## 5.2 HOW TO MOST EFFECTIVELY USE SYNTHETIC DATA FOR CONTINUAL LEARNING?

We empirically evaluate how to more effectively use synthetic data for continual learning, by comparing our method G2D to our generative replay alternative. We use the same generator (i.e., Stable Diffusion or T5) and classifier architectures (ViT B-16 or BERT-base) as G2D, and follow the standard generative replay approach (Shin et al., 2017): At domain $D_t$, we finetune the generator $G$ with samples from the current domain, sample from $G$, and add the generated samples to a replay buffer. At domain $D_{t+1}$, we sequentially train our classifier on the union of synthetic samples from domains $1, ..., t-1$ and real samples from domain $t$. We evaluate this generative replay implementation on all considered benchmarks for comprehensiveness. For the image domain, we show that across all benchmarks (in Tables 1, 2, and 3), utilizing the same synthetic samples for training a domain discriminator brings an absolute performance boost of 5.5, 2.83, and 8.7 points, respectively, over using the synthetic samples for finetuning the downstream classifier. On the question-answering bench-

| Method | ROC AUC (↑) |
|---|---|
| ER (50/class) | 84.94 ± 0.69 |
| EWC | 72.29 ± 1.71 |
| L2P | 81.88 ± 0.48 |
| S-Prompts | 79.38 ± 0.12 |
| Supervised Contrastive - CaSSLe | 73.45 ± 0.70 |
| Generative Replay (Ours) | 81.29 ± 0.57 |
| **G2D** | 89.14 ± 2.47 |
| **G2D (Full FT)** | 89.95 ± 1.47 |
| Upper Bound (MTL) | 91.56 ± 0.56 |

Table 3: Results on DermCL. Comparing performance in terms of average ROC AUC after training on the last domain (averaged over 5 random seeds). L2P and G2D use a buffer size of 0 images per class. ↑ indicates higher is better.

| Method | Average $F_1$(↑) |
|---|---|
| ER | 61.2 ± 1.8 |
| MbPA++ | 61.9 ± 0.2[†] |
| Meta-MbPA | 64.9 ± 0.3[†] |
| SeqFT | 56.6 ± 5.7 |
| EWC | 55.9 ± 3.7 |
| Generative Replay (Ours) | 58.5 ± 3.7 |
| **G2D** | 64.7 ± 0.2 |
| **G2D (Full FT)** | 66.6 ± 0.7 |
| Upper Bound (MTL) | 68.6 ± 0.0 |

Table 4: Results on Question Answering task. Comparing performance in terms of average $F_1$ across methods after training on the last domain (averaged over 4 random domain sequences). ↑ indicates higher is better, † denotes results obtained from Wang et al. (2020). ER, MbPA++ and Meta-MbPA use a buffer size of 1% actual samples. Our approach demonstrates competitive performance, even in the absence of retaining the actual samples, when compared to state-of-the-art methods.

mark, our method improves over generative replay by 6.2 $F_1$ points. Note for generative replay, full fine-tuning leads to greater performance than the LoRA fine-tuning approach; thus we report the former for optimal performance of the baseline (see Appendix C.3 for details). In Table 8 (see Appendix F), we visualize generated samples for their quality. These gains can be attributed to the fact

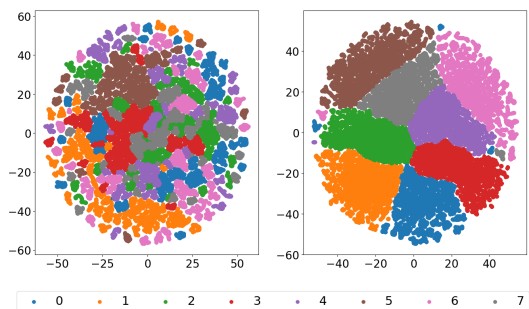

Figure 2: Domain discrimination visualization. t-SNE visualizations of domain clusterings for CORe50 benchmark (8 domains in total). On the left, the clusterings for the S-Prompts (Wang et al., 2022a) domain identification method is visualized. On the right, the clusterings for our domain discriminator is visualized.

that marginal distribution over a given domain $p(x)$ can be modeled more easily than the conditional $p(y|x)$. *Therefore, we observe that synthetic samples used for domain discrimination bring greater performance improvement than those used for downstream classification.*

### 5.3 DOMAIN DISCRIMINATION ANALYSIS

We analyze our domain discriminator, by comparison with the following methods. For the image domain, we compare it with a previous state-of-the-art domain incremental learning method, S-Prompts (Wang et al., 2022a). S-Prompts utilizes K-Means during training to store centroids for each domain and employ K-NN during inference to identify the domain of a given test image feature by determining its nearest centroid. Notably, both K-Means and K-NN operations are conducted in the feature space of the fixed pre-trained vision transformer. In Figure 2, we present t-SNE plots (Van der Maaten & Hinton, 2008) illustrating domain clusterings for the CORe50 benchmark, which comprises a total of eight domains. These plots compare the performance of the S-Prompts domain identification method with our discriminator. It is evident that our method achieves superior clustering and excels in domain identification (achieving $98.48\%$ accuracy). Figure 3 displays t-SNE plots for the text domain, visualizing the domain discriminative capability (Aharoni & Goldberg, 2020) of a pre-trained language model with a discriminator trained on synthetic samples. Notably, there is confusion between the TrWeb (orange) and TrWiki (green) domains, both derived from the same TriviaQA dataset (Joshi et al., 2017). Similarly, the TrWiki (green) and SQuAD (red) domains, originating from the same Wikipedia source, necessitate explicit discriminator training. It is evident that training an explicit discriminator results in superior clustering. To assess domain identification performance using generated samples (achieving $94.5\%$ accuracy), we compare it with a theoretical upper bound, namely a discriminator trained using real samples (achieving $97.1\%$ accuracy). The results show very close performance, with similar clustering patterns. *In summary, it is evident that our method significantly improves domain identifiability for both modalities.*

### 5.4 PARAMETER EFFICIENT FINE-TUNING ANALYSIS

Although we focus on high-stakes settings (e.g., healthcare), where optimizing performance is often most important, we employ parameter-efficient fine-tuning methods to address potential efficiency concerns. For the vision domain generator, we fine-tune only the LoRA weight matrices (Hu et al., 2021) added to the attention layers of the frozen pre-trained backbone. For the text domain, we use prompt tuning (Lester et al., 2021) to learn parameter-efficient generative models. Despite this great reduction in parameters, we observe that the performance of our domain discriminator trained on these generated samples is sufficient for domain identification and outperforms previous existing methods. For our downstream classifiers, we fine-tune only $1.04 \sim 2.5\%$ of trainable parameters. We further analyze a full fine-tuning variant of our method, i.e., G2D (Full FT), to study the potential performance drop we are experiencing from the great reduction in parameters. *We find that our method is comparable or at times has better performance than the naive full fine-tuning variant* (see Tables 1, 2, 3, and 4).

## 6 CONCLUSION

In this work, we investigate a novel approach to leveraging generative models for continual learning. We demonstrate its effectiveness across established vision and language benchmarks and a new, challenging dermatology imaging task, achieving improvements of 7.6, 6.0, 6.2, and 10.0 points over prior state-of-the-art domain incremental learning approaches. Further, we analyze how to most effectively leverage the capabilities of generative models and synthetic data for continual learning, by comparing our method to generative replay. Surprisingly, we find that training a domain identifier is more effective than using the *same* synthetic samples to augment training data for downstream classification. We further analyze our domain discriminator, by comparing it with previous domain discrimination approaches, unsupervised clustering methods, and a discriminator trained using real samples, where we find that our method significantly improves domain identifiability for both modalities.

## Reproducibility Statement

We introduce a novel benchmark DermCL constructed from publicly available dermoscopic image datasets – HAM10000 (Tschandl et al., 2018), BCN2000 (Cassidy et al., 2022), PAD-UEFS-20 (Pacheco et al., 2020), and DDI (Daneshjou et al., 2022). Following previous practice (e.g., DomainNet), we keep the domain sequence fixed for consistency to HAM10000 → BCN2000 → PAD-UEFS-20 → DDI and report the average performance over multiple random seeds. Further, in §4.3 and Appendix C, we detail all hyper-parameters to enable the reproducibility of all our experiments. We also plan to release our code upon publication.

## Ethics Statement

Training large models is expensive and has a detrimental impact on the environment (Strubell et al., 2019). Continual learning on top of existing models is cheaper and better compared to re-training from scratch since it requires a much smaller number of steps. With G2D, we aim to reduce the need to re-train models from scratch whenever a new set of data is added is encountered thereby making it cheaper and better for the environment. Furthermore, we implement our proposed method with parameter-efficient fine-tuning techniques, as aforementioned in §5.4.

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

| Method | Average Accuracy (↑) |
|--------|---------------------|
| S-Prompts | $83.13 \pm 0.51^{\dagger}$ |
| **G2D** | $89.11 \pm 0.30$ |
| **G2D (ContDD)** | $88.13 \pm 0.01$ |

Table 5: Ablation results on CORe50: G2D (ContDD) is a variant of our method where the domain discriminator is also trained in a continual manner. In the table, we include G2D as well as S-Prompts (i.e., the previous SOTA method) for easy comparison for the reader. We compare performance in terms of average accuracy after training on the last domain (averaged over 5 random seeds). ↑ indicates higher is better and † denotes results obtained from Wang et al. (2022a).

## A  DATASETS AND ORDERINGS

**Question Answering.**  Our processed dataset includes SQuAD with $90,000$ training and $10,000$ validation examples, TriviaQA (Web) with $76,000$ training and $10,000$ validation examples, TriviaQA (Wikipedia) with $60,000$ training and $8,000$ validation examples and QuAC with $80,000$ training and $7,000$ validation examples. We consider the following dataset orders for question answering:
i. QuAC→TrWeb→TrWik→SQuAD
ii. SQuAD→TrWik→QuAC→TrWeb
iii. TrWeb→TrWik→SQuAD→QuAC
iv. TrWik→QuAC→TrWeb→SQuAD

**DermCL.**  Conducting classification on dermoscopic images presents complexities arising from intraclass variations encompassing lesion texture, scale, and color. This benchmark offers a sequence of four dermatology imaging tasks. Distribution shifts are present across all four domains (HAM10000, BCN2000, PAD-UEFS-20, and, DDI), in both demographics and data collection techniques. BCN2000 dataset was collected from Spanish hospitals between 2010 and 2016, PAD-UEFS-20 dataset was obtained from Brazilian hospitals in 2020, and HAM10000 dataset was gathered over the past 20 years from hospitals in Austria and Australia. Dermatoscopes were used for collecting images in BCN2000 and HAM10000, while smartphone cameras were utilized for PAD-UFES-20. Lastly, Diverse Dermatology Images (DDI) is a biopsy-proven skin disease dataset with diverse skin tone representation. DDI has been attributed to exhibiting a huge performance drop, due to the presence of more dark skin tones and uncommon diseases. The label space for DermCL is defined as the following 5 unified labels: MEL, NEV, BCC, AKIEC, and Other diseases. All four datasets in the sequence are publicly available.

## B  ABLATION: CLASS INCREMENTAL LEARNING CHALLENGE OF DOMAIN DISCRIMINATOR

We include the following ablation to study the performance of when the domain discriminator is also trained in a continual manner, termed G2D (ContDD). Note that this turns the learning of domain discriminator into a class-incremental continual learning problem, introducing a new challenge on top of our original domain incremental learning problem. To assess this, we select the benchmark dataset with the most number of domains, which is the CORe50 benchmark with a total of 8 different domains. We find that this continual fine-tuning of the domain discriminator results in a performance drop of less than 1 point (see Table 6). Overall, our conclusions remain the same in that we still outperform existing state-of-the-art baselines by a substantial margin.

## C  IMPLEMENTATION DETAILS

### C.1  IMAGE EXPERIMENTS

In our method, we use Stable Diffusion (Rombach et al., 2022) as our conditional diffusion model, with weights from the CompVis/stable-diffusion-v1-4 checkpoint. We finetune for 250,000 total

steps with a learning rate of $\alpha = 1e - 5$, using the LoRA implementation. For the downstream classification task, we use a ViT-B/16 backbone (Dosovitskiy et al., 2020), pretrained on ImageNet-1K (Russakovsky et al., 2015). We implement baselines with the same architecture and pre-trained checkpoints, for consistency. For hyperparameter search, we use the source hold-out performance to select the best combination of parameters. For each dataset, we perform a sweep over different combinations of learning rate $\alpha \in [1e - 4, 5e - 4, 1e - 3, 5e - 3, 1e - 2, 5e - 2, 0.01]$, and batch size $\in [64, 128, 256]$. We use default hyperparameters for LoRA, resulting in the rank ($r$) of 16 and scaling factor (i.e., lora alpha) of 16. To ensure that we are evaluating our baselines comprehensively, we also run a hyperparameter search for different regularization values $\lambda \in [0.5, 1, 10, 100]$ for the Elastic Weight Consolidation (EWC) method. For DomainNet, we train for $20 \sim 30$ epochs. For CORe50, we train for 10 epochs. and For DermCL, we train for 10 epochs.

### C.2   Text experiments

For our generator, we use the prompt tuning to learn the parameter-efficient models (Lester et al., 2021). We use the pre-trained T5-Large v1.1 checkpoint adapted for prompt tuning as the backbone (Raffel et al., 2020) and the prompt embeddings are initialized randomly. We set the prompt length to 400 tokens which accounts for 819K trainable parameters, i.e., around 0.1% in comparison to 784M frozen T5-Large parameters. We input a special token into the model and conditionally generate a document content, question, and answer, all separated by the special tokens. We employ the Adam optimizer (Kingma & Ba, 2014) with a learning rate of 1.0, a warmup ratio of 0.01, and linearly decay the learning rate over 5 epochs, use a batch size of 8 and set weight decay to $1e^{-5}$. Our maximum sequence length is set to 512, and we truncate the document content after tokenizing the question-answer pair. During the generation process, we provide multiple text prompts. We use the following text prompts to conditional generate synthetic samples – "Generate article, question and answer.", "Generate context, question and answer.", "Generate answers by copying from the generated article.", "Generate factual questions from the generated article." During generation, we use ancestral sampling, which selects the next token randomly based on the model's probability distribution over the entire vocabulary, thereby reducing the risk of repetition. We generate samples with a minimum length of 50 tokens and a maximum of 1,000 tokens, retaining only those samples that contain exactly one question-answer pair with the answer included in the generated document content.

For training our domain discriminator and expert, we use the low-rank adaptation (LoRA; Hu et al., 2021) and freeze the pre-trained BERT-Base (Kenton & Toutanova, 2019) backbone. BERT-base has 12 Transformer layers, 12 self-attention heads, and 768 hidden dimensions (110M parameters). We train our discriminator for 5 epochs and expert for 3 epochs using Adam optimizer and the learning rate is set to 5e-4. For LoRA, we set the dimension of the low-rank matrices (r) to 32 and the scaling factor to 32, which gives us around 1.2M trainable parameters (1.07% of full BERT-Base 110M parameters). In the case of a full fine-tuning scenario, for training our expert, we mainly set hyper-parameters as mentioned in de Masson D'Autume et al. (2019). We use Adam as our optimizer, set dropout to 0.1, and the base learning rate to 3e-5. We use a training batch of size 8, set the maximum total input sequence length after tokenization to 384 and to deal with longer documents we set document stride to 128. We also set the maximum question length to 64. The hyper-parameters for baseline methods are set as described in Wang et al. (2020). For ER (and Generative Replay) we retain (or sample) 1% examples which account for around 6,000 examples across all four considered domains

### C.3   Further Implementation Details of Generative Replay

We implement Generative Replay following standard practice (Shin et al., 2017), where the downstream classifier is continually fine-tuned on the union of synthetic images from previous domains and real data from the current domain (i.e., all tasks so far are given equal weight). More explicitly, if we have seen $t - 1$ tasks so far and currently training on the $t$-th task, then we consider all $t$ tasks with equal weight $\frac{1}{t}$. This aligns with the manner in which the domain discriminator for G2D is trained.

| Method | DomainNet | CORe50 |
|---|---|---|
| Generative Replay (Full FT) | $52.97 \pm 0.07$ | $86.28 \pm 0.55$ |
| Generative Replay (LoRA) | $49.67 \pm 0.21$ | $83.64 \pm 0.45$ |
| Generative Replay (LoRA - Matched) | $50.12 \pm 0.09$ | $84.21 \pm 0.46$ |
| G2D (Full FT) | $54.37 \pm 0.90$ | $86.60 \pm 0.28$ |
| G2D (LoRA) | $\mathbf{58.45 \pm 0.56}$ | $\mathbf{89.11 \pm 0.30}$ |

Table 6: Generative Replay LoRA ablation results on DomainNet and CORe50. In the table, we include Generative Replay (Full FT) and both variants of G2D for easy comparison for the reader. Further experimental details are noted in Appendix C.3. The highest performance is bolded.

We sought to make the comparison with G2D fair by (a) using the same replay buffer (i.e., fixing to the same set of synthetic images / or text samples) for both methods and (b) using the "same" fine-tuning approach for classifier models (detailed as follows):

*Both* approaches (for fine-tuning classifier models of G2D and the classifier for Generative Replay) are done by sequential fine-tuning from the previous domain checkpoint. Hyperparameter tuning is done over the same set of hyperparameters (learning rates, batch size, etc.) and chosen based on performance on the held-out validation set of the first domain. Hyperparameters remain fixed throughout the domain sequence, for both approaches (see Appendix C.1 and C.2 for hyperparameter details). For G2D, we employ parameter-efficient fine-tuning (LoRA; Hu et al., 2021) while for G2D (Full FT), we employ full fine-tuning.

Note that for Generative Replay, we opted against employing parameter-efficient fine-tuning (LoRA; Hu et al., 2021) techniques due to its inferior performance compared to full fine-tuning (see Table 6). In the case of G2D, for two out of four benchmarks (DomainNet, CORe50), we see that LoRA does not hurt performance for fine-tuning the classifier, but rather results in slight performance improvement over full fine-tuning. We hypothesize that in the case of G2D, fine-tuning a smaller set of parameters is more tractable and perhaps leads to less overfitting, since we are using distinct sets of weights for simpler tasks (i.e., classification for a specific domain), relative to the more challenging task of training a common classifier for Generative Replay (i.e., classification on all seen domains). We run Generative Replay (LoRA) and Generative Replay (LoRA - Matched) ablations for the two aforementioned benchmarks as follows.

For Generative Replay (LoRA), we use the same LoRA hyperparameters (rank = 16) as we did in G2D (LoRA), resulting in the same number of trainable parameters 1.29% for both DomainNet and COre50. For comprehensiveness, we also ran experiments where we "match" the number of total trainable parameters, which we term Generative Replay (LoRA - Matched). For instance, for DomainNet, we have 6 domains resulting in 6 expert classifiers each with 1.29% trainable parameters. Thus, in terms of total LoRA parameters, there is a discrepancy between (1) sum of expert classifiers' LoRA parameters for G2D and (2) single classifier's LoRA parameters for Generative Replay. To account for this discrepancy, we adjust the rank hyperparameter to 16 x num_expert_classifiers to match the number of total parameters. This results in the exact same number of trainable LoRA parameters across both methods, resulting in using rank=(16x6)=96 for DomainNet and rank=(16x8)=128 for CORe50.

We can clearly observe that even after carefully matching the number of total trainable parameters, we *still* observe a performance drop when using LoRA for Generative Replay. We hypothesize that using LoRA more often hurts performance for Generative Replay as the classifier here has to perform the more challenging task of classification on *all seen domains*. On the contrary, for G2D, each classifier expert has a much simpler task of classification for *a specific domain*.

We note that if the reader wishes to do an apples-to-apples comparison in terms of fine-tuning techniques, we point the reader to comparing (i) G2D (Full FT) vs. Generative Replay (Full FT) and (ii) G2D (LoRA) vs. Generative Replay (LoRA) vs. Generative Replay (LoRA-Matched).

Based on the direct comparison in Table 6 of all variants (Full FT, LoRA) for both methods (Generative Replay, G2D), we highlight benefits of $0.32 \sim 8.66\%$ improvement for the full fine-tuning setting and $4.9 \sim 8.78\%$ improvement for the LoRA fine-tuning setting, which has additional benefits of parameter efficiency. Further, our findings highlight how to better utilize recent PEFT methods

in continual learning, in that when learning *independent* weights, performance is often at least on-par with full fine-tuning. On the other hand, learning one set of parameters for multiple tasks can lead to performance drops due to this restriction of parameters; therefore, not being able to utilize PEFT solutions in an optimal fashion. Understanding this phenomena more in detail is an interesting direction for future work.

## C.4    COMPUTATIONAL COST

We elaborate on details regarding computational cost of our method, relative to naive expert learning: Our method incurs an additional total of 6-8 hours of computational cost on a single A6000 GPU, due to fine-tuning and sampling from the generator. Training the task discriminator, which takes less than 1-3 hours is done in parallel with training the expert classifier, so given that we permit the use of one more GPU, it does not incur additional compute time. There is an accuracy and compute tradeoff: This increase in compute cost, results in a substantial performance improvement - up to an absolute 8.1 point increase compared to expert learning (see Table 4). In high-stakes applications such as healthcare, where missing even a single positive case (e.g., a fatal disease) could have critical consequences (e.g., a patient's death), this performance gain is significant, and thus, we deemed this amount of tradeoff is meaningful.

## C.5    LORA FINE-TUNING VS. FULL FINE-TUNING FOR GENERATOR

For parameter efficiency, we finetune our generator with low-rank adaptation (i.e., LoRA) (Hu et al., 2021). This implementation choice was decided by preliminary results, which demonstrated minor performance drops, compared to significant gains in terms of parameter efficiency (see Table 7).

| Method | Classification Accuracy Score (CAS) (↑) |
|---|---|
| Full Fine-tuning | $66.28 \pm 0.31$ |
| LoRA Fine-tuning | $64.17 \pm 0.20$ |

Table 7: Ablation results on DomainNet: We compare the Classification Accuracy Score (CAS) (Ravuri & Vinyals, 2019) for full fine-tuning of the generator vs. LoRA fine-tuning of the generator. Full fine-tuning the generator achieves a CAS of $66.28 \pm 0.31$, while LoRA fine-tuning reaches $64.17 \pm 0.20$, with only fine-tuning at most 2.5% of trainable parameters. Noting this performance vs parameter efficiency tradeoff, we proceeded with the LoRA-based implementation choice to make the synthetic data generation process parameter efficient.

## D    ONGOING DISCUSSION ON UTILITY OF GENERATIVE MODELS IN HEALTHCARE DEPLOYMENT SETTINGS.

In general, *and for good reason*, practice in healthcare moves considerably slower than exploratory machine learning research. It is generally the case that ideas take root in the research community long before they show up in the clinic. Following this convention, due to the recency of successes of generative models (relative to discriminative models), contractual or regulatory requirements surrounding generative models is still in nascent stages of development.

*What is the current status quo?* The setting where model weights may be shared but not the actual training data is a well-known setting in the healthcare domain (Kamran et al., 2022; Ulloa-Cerna et al., 2022; Walsh et al., 2023). We elaborate on two examples: (1) Kamran et al. (2022) presents a multisite external validation study for early identification of COVID-19 patients at risk of clinical deterioration, which require sharing the model trained on private EHR data from one US hospital with 12 other US medical centers; (2) Ulloa-Cerna et al. (2022) presents a multisite external validation study for model development for identifying patients at increased risk of undiagnosed structural heart disease, which requires sharing the model trained on private EHR data and patient echocardiography reports from one site with 10 other independent sites. While these are generally examples of discriminative models being shared across facilities as opposed to generative models, this demonstrates the general principle that in such domains, **model sharing is often permissible in settings where data sharing is not.**

On one hand, it seems intuitive that healthcare institutions might be queasier about sharing generative models than sharing discriminative models. On the other hand,

- Healthcare institutions are even queasier about sharing real data — and to this end there is a **large mainstream line of work investigating the use of generative models for direct sharing** or for producing synthetic datasets that could be disseminated in lieu of actual patient data (Chen et al., 2021; Coyner et al., 2022; DuMont Schütte et al., 2021)

- From a standpoint of most contractual or regulatory requirements, it is not yet clear even if generative models sit in a different category than discriminative models or if they should follow the same current regulatory requirements for discriminative models.

- **How institutional practices develop and the regulatory environment evolve** will be informed, to a large degree, by exploratory research that characterizes both (i) the potential benefits and (ii) the potential risks associated with the dissemination of generative models trained on medical data. **We see our research as helping to elucidate the potential benefits.**

# E    DOMAIN DISCRIMINATION CLUSTERINGS

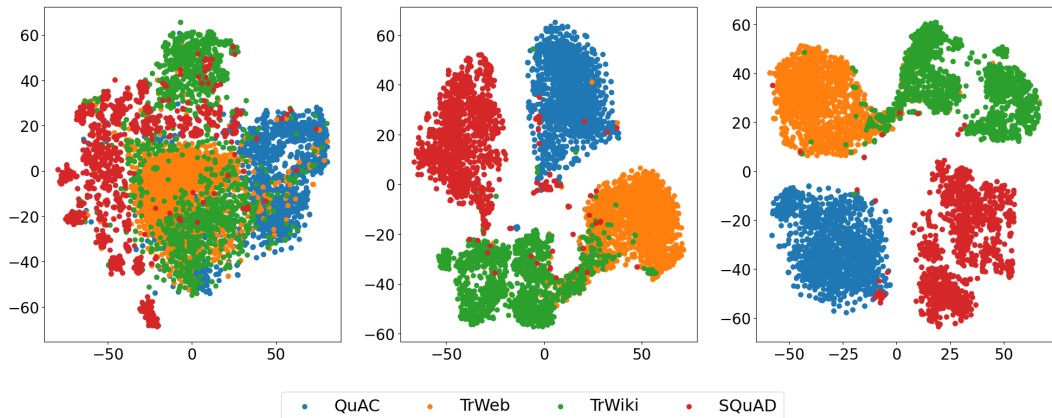

Figure 3: Domain discrimination visualization. t-SNE visualizations of domain clusterings for question-answering benchmark (4 domains in total). The left plot highlights the implicit domain discriminative nature of pre-trained BERT-Base language model representations (Kenton & Toutanova, 2019). Notably, there is confusion between the TrWeb (orange) and TrWiki (green) domains, both derived from the same TriviaQA dataset. Similarly, the TrWiki (green) and SQuAD (red) domains, originating from the same Wikipedia source, necessitate explicit discriminator training. In the middle plot, we visualize the clustering of representations from the discriminator trained using generated samples, achieving a domain discrimination accuracy of $94.5\%$. On the right plot, we present the clustering from the discriminator trained using real samples, with an accuracy of $97.1\%$. Remarkably, the discriminator trained with synthetic samples closely mirrors the performance and clustering patterns of the discriminator trained using real data.

# F    EXAMPLE GENERATIONS

We include example generations for both image and text domains.

## F.1    IMAGE DOMAIN

## F.2    TEXT DOMAIN

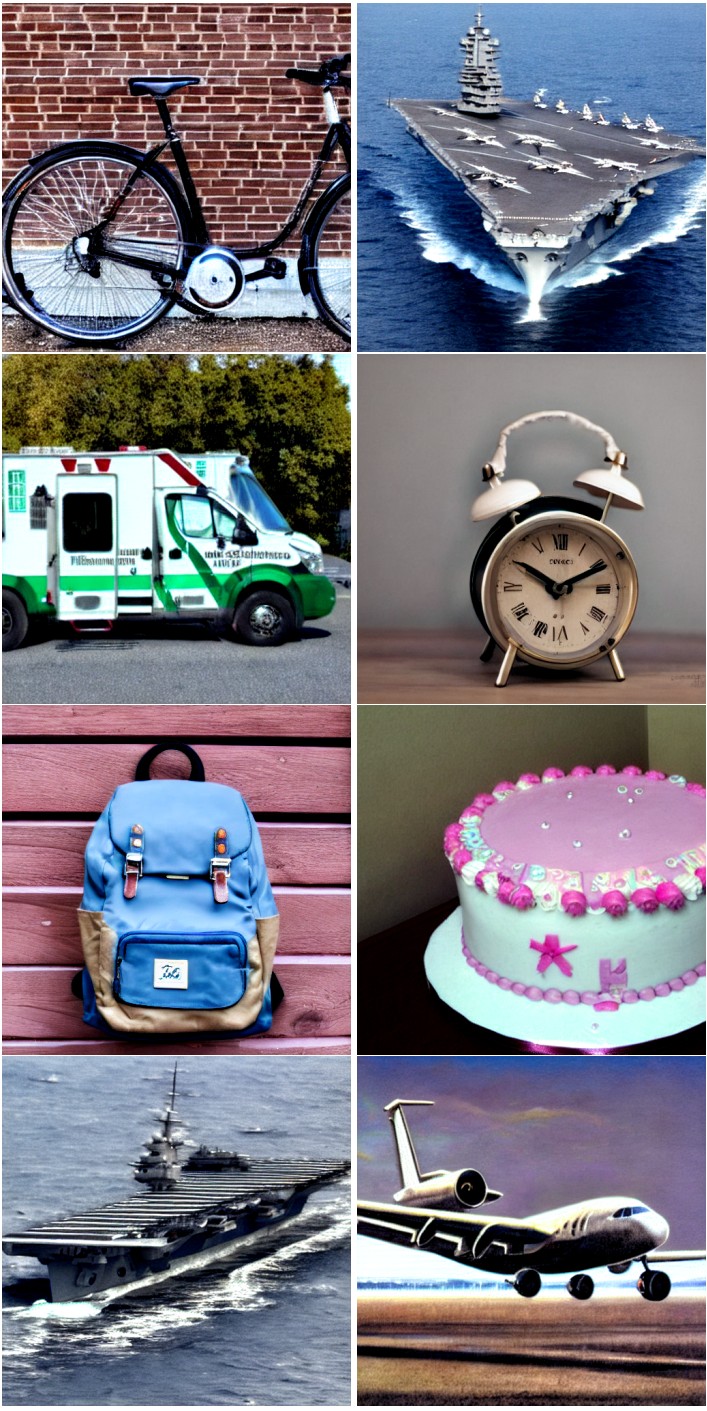

Figure 4: Examples of *generated* images from DomainNet.

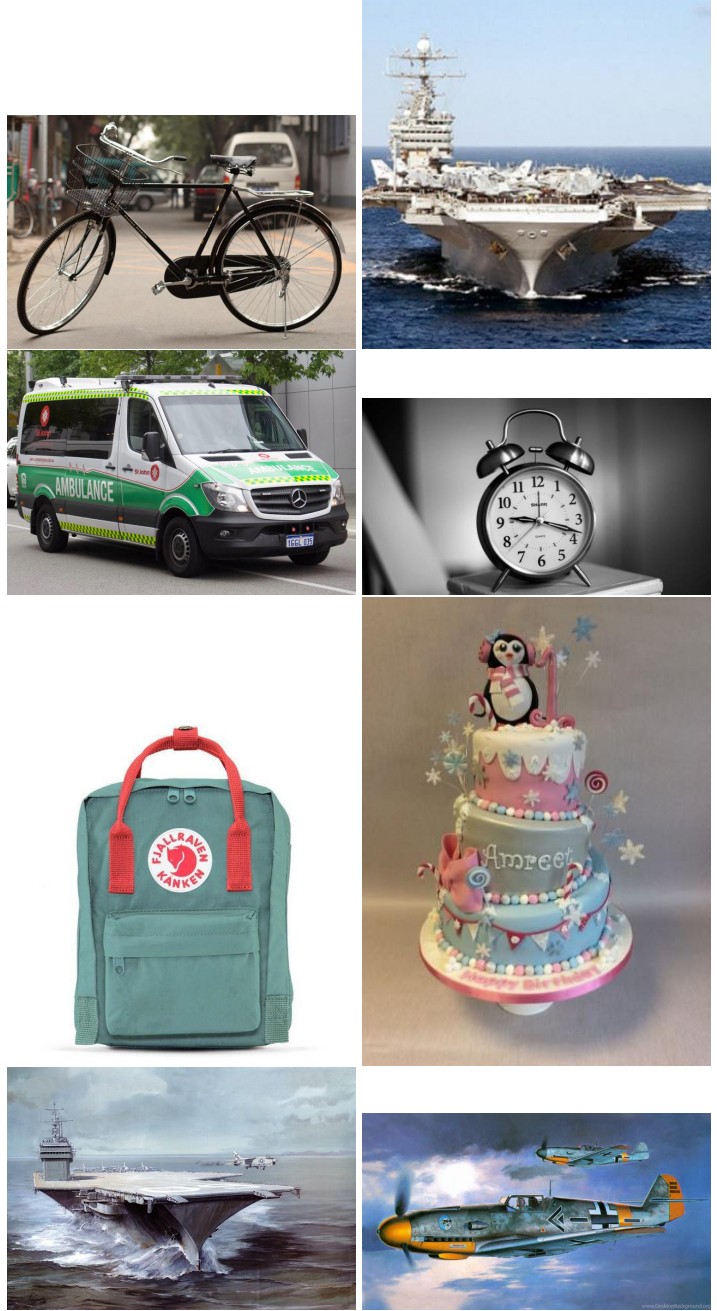

Figure 5: Examples of *real* images from DomainNet (for comparison).

| Dataset | Fields | Generated sample |
|---|---|---|
| SQuAD | | **Correct** |
| | *Context:* | During the late 19th and early 20th centuries, the city's trade sector expanded greatly, and through the 20th century, more than half of its residents worked toward higher status in the military. Other industries included industry, commerce, public administration, and medicine. Its largest sector was public services - police, fire services, and healthcare and was the nation's third largest. |
| | *Question:* | What is one of the industry sectors that were the biggest? |
| | *Answer:* | public services |
| | | **Incorrect** |
| | *Context:* | In the United Kingdom there is a general agreement between the government and the private sector in principle that both private and publicly funded institutions of higher education constitute university colleges. Further, there is a mutual agreement between the independent college and the university to promote higher education. However, in both cases all the institutions of higher education are either controlled by private individuals or by a national agency, in such a way as to protect freedom of expression. |
| | *Question:* | What are some of the institutions of higher education that are controlled by private individuals? |
| | *Answer:* | private individuals |

Table 8: Generated samples (context, question-answer pair) for the SQuAD domain. For the incorrectly generated samples, we underline one possible correct answer.

