# OpenReview forum: "Generate to Discriminate: Expert Routing for Continual Learning"
_ICLR.cc/2024/Conference — Submitted to ICLR 2024_

### Official Review · Reviewer_833z · 2023-10-29

**Soundness:** 3 good
**Presentation:** 2 fair
**Contribution:** 3 good
**Rating:** 6
**Confidence:** 3

**Summary:**

- The authors suggest a new method for domain-incremental continual learning, leveraging recent approaches in conditional generative models. Specifically, the authors generated samples to train a domain discriminator which, in turn, is used as expert gate, to route samples at inference time to the appropriate expert model.
- Furthermore, the paper suggests a new benchmark dataset for domain-incremental learning, named DermCL, combining different dermatologic datasets.
- They evaluate their approach on 3 vision and 1 text (QA) tasks.

**Strengths:**

- The paper addresses a relevant topic, namely domain-incremental catastrophic forgetting.
- The approach follows a simple and neat idea, which is to employ generated samples to train a gate model, instead of using them in an augmentation step to finetune the classification model.
- The authors provide a good summary of related work.
- The authors conduct extensive experiments with various datasets and multiple modalities (image, text).

**Weaknesses:**

- It is hard to connect the table with the text -> e.g. in tab 1: where is ER? Whats CaSSLe? What’s the difference between G2D and G2D (Full FT) –> roughly explained much later in the text? Why are different methods compared for different (vision) datasets?

**Questions:**

- Will the new benchmark dataset be published?

---

> ### Author Response · Authors · 2023-11-16
> **Response to Reviewer 833z**
>
> Thank you very much for your time and thoughtful feedback! We have now made the following changes which we believe address all your concerns and strengthen the paper
> * Added additional results on ER (50/class) for DomainNet and DermCL (see Tables 1,3)
> * Added additional results on CaSSLe for CORe50 and DermCL (see Tables 2,3)
> * Added clarification to Table 1 caption on the difference between G2D vs. G2D (Full FT)
>
> > In tab 1: where is ER? Whats CaSSLe?...Why are different methods compared for different (vision) datasets
>
> Thank you for bringing this to our attention. As per your suggestion for completeness, we have run ER and CaSSLe numbers for all remaining baselines and have updated Tables 1,2,3 accordingly, which now unifies all baselines assessed for the vision datasets. TLDR: Our method outbeats both baselines across all datasets by a large margin. We have included a summary of the new results in the table below (*denotes results obtained from [1]):
>
> | Method                                | DomainNet         | CORe50           | DermCL           |
> |---------------------------------------|-------------------|------------------|------------------|
> | ER (50/class)                         | 52.79 ± 0.03      | 80.10 ± 0.56*   | 84.94 ± 0.69     |
> | Supervised Contrastive - CaSSLe       | 50.90*           | 75.68 ± 0.60     | 73.45 ± 0.70     |
> | G2D                                   | 58.45 ± 0.56      | 89.11 ± 0.30     | 89.14 ± 2.47     |
>
> For further clarification: In our earlier version, these two baselines were only evaluated on selected datasets because previous works [1] [2] included ER and CaSSLe results only on selected datasets and we directly obtained those results. CaSSLe [2] is a method for continual self-supervised learning and ER cannot be performed in our setup due to data sharing constraints; and thus, we had not focused too much on evaluation of these two baselines in our earlier draft. As noted, we have now updated our draft to address your concern.
>
> >  What’s the difference between G2D and G2D (Full FT) –> roughly explained much later in the text?
>
> We apologize for any confusion, due to the description coming much later in the text. We have added this clarification in the caption of Table 1, where G2D (Full FT) is first mentioned. We further elaborate on the difference here:
> The difference between G2D and G2D (Full FT) is that G2D is using parameter-efficient approaches for fine-tuning, while G2D (Full FT) is a variant of our method that uses full fine-tuning of all parameters. For G2D, our classifier models are fine-tuned using LoRA [3], where low-rank “update matrices” are added to the query and value matrices of the attention blocks of the base model. During fine-tuning, only these matrices are trained, while the original model parameters are kept frozen. At inference time, the update matrices are merged with the original model parameters to produce the final output.
>
> >  Will the new benchmark dataset be published?
>
> Yes, we plan to publicly release the newly proposed benchmark dataset DermCL along with all code associated with this work.
>
> We hope these answers and additional experiments help clarify your questions. We would be happy to answer any further questions you have. If you do not have any further questions, we hope that you may consider raising your score. Thank you again for your constructive feedback!
>
> **References**
>
> [1] S-Prompts Learning with Pre-trained Transformers. Wang et. al 2023
>
> [2] Self-Supervised Models are Continual Learners. Fini et. al 2022
>
> [3] Lora: Low-rank adaptation of large language models. Hu et. al 2021

---

> > ### Author Response · Authors · 2023-11-20
> > **Follow-up to Reviewer 833z**
> >
> > Thank you very much for your time and thoughtful feedback! We hope our answers and additional experiments above help clarify your questions. As the author-reviewer discussion deadline is approaching soon, we wanted to check if you have any further questions. If you do not have any further questions, we hope that you may consider raising your score. Thank you again for your constructive feedback!

---

> > > ### Author Response · Authors · 2023-11-22
> > > **Any further questions?**
> > >
> > > Thank you again for your comprehensive and useful review! According to your suggestions, we have
> > >
> > > 1. Added additional results on ER for DomainNet and DermCL (see Tables 1,3)
> > > 2. Added additional results on CaSSLe for CORe50 and DermCL (see Tables 2,3)
> > > 3. Added clarifications to Table 1 caption on the difference between G2D vs. G2D (Full FT)
> > >
> > > Do you have any further questions? If you have any more suggestions or feedback, please let us know

---

### Official Review · Reviewer_YtsH · 2023-10-30

**Soundness:** 2 fair
**Presentation:** 3 good
**Contribution:** 3 good
**Rating:** 5
**Confidence:** 3

**Summary:**

This paper proposed to address the domain-incremental learning problem by learning domain-specific models combined with a model capable of distinguishing between domains. This domain discriminator is trained using synthetic data from a continually fine-tuned generative model. An important empirical demonstration is that the authors show that this indirect approach (i.e., first identify the domain, then solve the problem) works better than when directly learning a model to solve the problem in all domains while replaying the same synthetic data sample.

**Strengths:**

I consider demonstrating that it can be more efficient to address a domain-incremental learning problem in an indirect way (i.e., G2D; first identify the domain, then solve the problem) than in a direct way (i.e., Generative Replay; directly learn to solve the problem in all domains) an important and insightful contribution.

**Weaknesses:**

Unfortunately, I think that the paper does not provide enough experimental details to properly assess whether the comparison between G2D and Generative Replay is performed in a fair manner. In particular, based on the provided details, it is unclear to me whether Generative Replay has been implemented in an optimal manner. Examples of details / explanations that should be provided:

- How is / are the classifier model(s) finetuned? In section 5.4. it is stated “we fine-tune only 1.04 ~ 2.5% of trainable parameters”. How was this percentage decided? How is it decided which parameters are fine-tuned? Is this approach of fine-tuning the same for the classifier models of G2D and the classifier model of Generative Replay?

- With Generative Replay, how are the loss on the replayed data and the loss on the data from the current task weighed? Are they simply added? Or are they balanced in such a way as to approximate the joint loss over all domains so far?


Could the authors explain why they took the S-iPrompts results from the Wang et al (2022a) paper, but not the S-liPrompts results?

On p5 towards the bottom the authors claim that ER with a limited buffer size is an upper bound for generative replay. This does not seem correct.

**Questions:**

Most importantly, the authors should provide full details regarding how the generative replay experiments were implemented in order for the reviewers to be able to judge whether the key comparison of this paper was performed in a fair manner.

Could the authors explain why they took the S-iPrompts results from the Wang et al (2022a) paper, but not the S-liPrompts results?

I would be happy to actively engage in the discussion period.

---

> ### Author Response · Authors · 2023-11-16
> **Response to Reviewer YtsH (1/2)**
>
> Thank you very much for your time and thoughtful feedback! We have now made the following changes which we believe address all your questions and strengthen the paper
>
> * Added a section on implementation details of Generative Replay in Appendix C.3
> * Added a summary of these clarifications to Section 5.2 with a reference to Appendix C.3
> * Updated section 4.2 to correct description about ER (see below)
>
>  > ... does not provide enough experimental details to properly assess whether the comparison between G2D and Generative Replay is performed in a fair manner…it is unclear to me whether Generative Replay has been implemented in an optimal manner
>
> We have updated the draft with the following clarifications (see Section 5.2  and Appendix C.3). Further, we plan to release all code regarding our implementation of both G2D and Generative Replay, for the community.
>
> We sought to make the comparison fair by (a) using the same replay buffer (i.e., fixing to the same set of synthetic data) for both methods and (b) using the *same* fine-tuning approach for classifier models (see below):
>
> > How is / are the classifier model(s) finetuned?...Is this approach of fine-tuning the same for the classifier models of G2D and the classifier model of Generative Replay?
>
> To be explicit, we will answer this question for (1) G2D (Full FT) vs. Generative Replay and (2) G2D vs. Generative Replay,
>
> 1. Comparison of G2D (Full FT) vs. Generative Replay:
>
> Yes, the approach remains the same. *Both* approaches (for fine-tuning classifier models of G2D (Full FT) and the classifier for Generative Replay) are done by:
> * Sequential fine-tuning from the previous domain checkpoint
> * Hyperparameter tuning is done over the same set of hyperparameters (learning rates, batch size, etc.) and chosen based on performance on the held-out validation set of the first domain. Hyperparameters remain fixed throughout the domain sequence, for both approaches (see Appendix C.1 and C.2 for hyperparameter details).
>
> 2. Comparison of G2D vs. Generative Replay:
>
> The fine-tuning procedure (above two bullet points) remains the same. There is just one difference: For G2D, we employed parameter-efficient fine-tuning techniques to improve the parameter efficiency aspect of our method:
> * G2D uses a parameter-efficient fine-tuning method (i.e., LoRA)
> * Generative Replay uses the same full fine-tuning method as G2D (Full FT).
>
> For generative replay, we opted against employing parameter-efficient fine-tuning (LoRA) techniques due to its inferior performance compared to full fine-tuning. The drawback arises from the sequential learning of multiple domains using limited trainable parameters in LoRA, which adversely impacts performance. Therefore, we presented the *optimal performance of Generative Replay*, achieved through the full fine-tuning procedure, mirroring the approach used in G2D (Full FT). We apologize that this might have not been clear. We have updated the draft to clarify this point (see Section 5.2 and Appendix C.3). Thank you for raising this clarification point!
>
> >  In section 5.4. it is stated “we fine-tune only 1.04 ~ 2.5% of trainable parameters”. How was this percentage decided? How is it decided which parameters are fine-tuned?
>
> The number of trainable parameters (so-called “update matrices”) and which of them are fine-tuned is controlled by (1) the dimension used by the LoRA update matrices (i.e., r) and (2) the scaling factor (i.e., lora alpha), which gives the flexibility to balance a trade-off between end performance and parameter efficiency [2]. For our vision experiments, we set the default values for both hyperparameters (r = 16, lora alpha = 16). For our text experiments, we set the rank and scaling factor to 32 to match the performance of parameter-efficient finetuning with full finetuning on a single domain. These parameter choices result in 1.04 ~ 2.5% of trainable parameters.
>
> > ... With Generative Replay, how are the loss on the replayed data and the loss on the data from the current task weighed? Are they simply added? Or are they balanced in such a way as to approximate the joint loss over all domains so far?
>
> Following standard practice [1], we simply add the loss, in an unbiased manner.

---

> ### Author Response · Authors · 2023-11-16
> **Response to Reviewer YtsH (2/2)**
>
> > Could the authors explain why they took the S-iPrompts results from the Wang et al (2022a) paper, but not the S-liPrompts results?
>
> As noted in Section 4.2 (end of third paragraph), for a fair comparison, we fix the model backbone of the downstream classifiers as ViT-B16 pretrained with ImageNet checkpoint. This is the same across all baselines and our methods. S-liPrompts uses a different pretrained backbone in the CLIP family [3]. Thus, we compare with the S-Prompts method (S-iPrompts), which uses the same pre-trained architecture.
>
> >  On p5 towards the bottom the authors claim that ER with a limited buffer size is an upper bound for generative replay. This does not seem correct.
>
> Thanks for catching this. We meant to clarify that Experience Replay (ER) with a lot of samples (not “limited buffer size”) is the upper bound. Indeed, Experience Replay (ER) with “limited buffer size” is not the upper bound. We have made this revision in the draft (see Section 4.2).
>
> We hope these responses helped clarify your questions. We would be happy to answer any further questions you have. If you do not have any further questions, we hope that you may consider raising your score. Thank you again for your constructive feedback!
>
> **References**
>
> [1] Continual Learning with Deep Generative Replay. Shin et. al 2017.
>
> [2] Lora: Low-rank adaptation of large language models. Hu et. al 2021.
>
> [3] S-Prompts Learning with Pre-trained Transformers. Wang et. al 2023.

---

> > ### Author Response · Authors · 2023-11-20
> > **Follow-up to Reviewer YtsH**
> >
> > Thank you very much for your time and thoughtful feedback! We hope our answers above help clarify your questions. As the author-reviewer discussion deadline is approaching soon, we wanted to check if you have any further questions. If you do not have any further questions, we hope that you may consider raising your score. Thank you again for your constructive feedback!

---

> > ### Comment · Reviewer_YtsH · 2023-11-21
> > **Response to author rebuttal**
> >
> > Thank you for the rebuttal and the clarifications.
> >
> > Based on these clarifications, I have the following comments:
> >
> > - I am surprised that using LoRA for the classifier used with Generative Replay does not help. The explanation given by the authors seems somewhat unlikely to me. If LoRA helps to improve performance when fine-tuning the classifier of G2D, and it helps to improve performance when continually training the generative model (for both G2D and Generative Replay; this is my understanding from subsection 4.3), it is surprising that it is unable to improve performance when continually training the classifier model for Generative Replay. I would say that this requires more explanation / demonstration.
> > - Considering the above issue, I now appreciate that the most direct comparison in the paper between G2D and Generative Replay is between “G2D (Full FT)” and “Generative Replay (ours)”. Based on this comparison, however, the results are substantially less clear / convincing.
> > - For Generative Replay, I am surprised by the authors’ choice to “simply add the loss” of the replayed data to the loss on the current task. This creates an imbalance between G2D and Generative Replay. As described in subsection 3.2, the domain discriminator of G2D is trained on the union of synthetic data from all tasks so far. That means that the domain discriminator is trained in a balanced way (i.e., all tasks so far are given equal weight). However, there is no such balance for Generative Replay. Because with Generative Replay, based on the new information provided by the authors, the loss of the classifier is constructed for 50% based on synthetic data from all past tasks and for 50% based on data from the current task. This means that for Generative Replay there is an imbalance (i.e., the last task is given higher weight than the previous tasks). I think it is likely this might negatively affect the performance of Generative Replay.

---

> ### Author Response · Authors · 2023-11-22
> **Response to Reviewer YtsH (1/3)**
>
> Thank you very much for your continuous engagement and useful review! We will try to address your questions below:
>
> > ...If LoRA helps to improve performance when fine-tuning the classifier of G2D, and it helps to improve performance when continually training the generative model…, it is surprising that it is unable to improve performance when continually training the classifier model for Generative Replay. I would say that this requires more explanation / demonstration.
>
> We have understood your points as follows: (i) if LoRA helps improve performance when fine-tuning the classifiers of G2D and (ii) if LoRA helps improve the performance when fine-tuning the generator (for both G2D and Generative Replay) (iii) then why does it not help for fine-tuning the classifier for Generative Replay.
>
> To address your concern, we have run two additional experiments: Generative Replay (LoRA) and Generative Replay (LoRA - Matched) (see Appendix C.3; also detailed below). First, we will address your first two points (i) and (ii). Then, we will move on to discuss our new results, which answers point (iii).
>
> First, on to point (ii), we apologize for the confusion and want to clarify that LoRA *does not help improve performance*, when fine-tuning the generator (for both G2D and Generative Replay) (see Appendix C.5). Our main reason to employ the LoRA approach for the generator was for parameter efficiency concerns, as noted in Section 4.3. We used the Classification Accuracy Score (CAS) [1], a widely employed metric to measure conditional generative models' quality, to assess performance of the generator. On DomainNet, full fine-tuning the generator achieves a CAS of 66.28 ± 0.31, while LoRA fine-tuning reaches 64.17 ± 0.20, with only fine-tuning at most 2.5% of trainable parameters (see Appendix C.5). Noting this performance vs parameter efficiency tradeoff, we proceeded with the LoRA-based implementation choice to make the synthetic data generation process parameter efficient,. We have added this clarification in Appendix C.5 and a footnote in Section 4.3 (due to current page limit constraints). Note that this LoRA-based implementation choice remains the same across both methods (Generative Replay and both variants of G2D), since we fix to the same set of synthetic data for fair comparison.
>
> As you note in point (i), for two out of four benchmarks (DomainNet, CORe50), we see that LoRA does not hurt performance for fine-tuning the classifier for G2D, but rather results in slight performance improvement over full fine-tuning. We hypothesize that in the case of G2D, fine-tuning a smaller set of parameters is more tractable and perhaps leads to less overfitting, since we are using distinct sets of weights for simpler tasks (i.e., classification for a specific domain), relative to the more challenging task of training a common classifier for Generative Replay (i.e., classification on all seen domains).
>
>
> For the aforementioned two benchmarks of concern, we have run the following additional experiments: Generative Replay (LoRA) and Generative Replay (LoRA - Matched). Highest performance is bolded.
>
>
> | **Method** | **DomainNet** | **CORe50** |
> |------------|---------------|------------|
> | Generative Replay (Full FT) | 52.97 ± 0.07 | 86.28 ± 0.55 |
> | Generative Replay (LoRA) | 49.67 ± 0.21 | 83.64 ± 0.45 |
> | Generative Replay (LoRA - Matched) | 50.12 ± 0.09 | 84.21 ± 0.46 |
> | G2D (Full FT) | 54.37 ± 0.90 | 86.60 ± 0.28 |
> | G2D (LoRA) | **58.45 ± 0.56** | **89.11 ± 0.30** |

---

> ### Author Response · Authors · 2023-11-22
> **Response to Reviewer YtsH (2/3)**
>
> As seen above, for direct comparison, we compare with all variants (Full FT, LoRA) for both methods (Generative Replay, G2D).
>
> For Generative Replay (LoRA), we use the same LoRA hyperparameters (rank = 16) as we did in G2D (LoRA), resulting in the same number of trainable parameters 1.29% for both DomainNet and CORe50. For comprehensiveness, we also ran experiments where we “match” the number of total trainable parameters, which we term Generative Replay (LoRA - Matched). For instance, for DomainNet, we have 6 domains resulting in 6 expert classifiers each with 1.29% trainable parameters. Thus, in terms of total LoRA parameters, there is a discrepancy between (1) sum of expert classifiers’ LoRA parameters for G2D and (2) single classifier’s LoRA parameters for Generative Replay. To account for this discrepancy, we adjust the rank hyperparameter (16 x num_expert_classifiers) to match the number of total parameters. This results in the exact same number of trainable LoRA parameters across both methods, resulting in using rank=(16x6)=96 for DomainNet and rank=(16x8)=128 for CORe50.
>
> We can clearly observe that even after carefully matching the number of total trainable parameters, we *still* observe a performance drop when using LoRA for Generative Replay. We hypothesize that using LoRA more often hurts performance for Generative Replay as the classifier here has to perform the more challenging task of classification on *all seen domains*. On the contrary, for G2D, each classifier expert has a much simpler task of classification for *a specific domain*. We have reflected these clarifications and additional experiments in the revised draft (see Appendix C.3).
>
> > Considering the above issue, I now appreciate that the most direct comparison in the paper between G2D and Generative Replay is between “G2D (Full FT)” and “Generative Replay (ours)”. Based on this comparison, however, the results are substantially less clear / convincing.
>
> As you note, we acknowledge that when comparing G2D (Full FT) vs. Generative Replay for two of the four benchmarks (DomainNet and CORe50), the performance gains are less than G2D vs. Generative Replay. As per your concern, we have reworded clarifications in Appendix C.3 to better reflect this and have run the additional experiments for the two benchmarks for direct comparison between Full FT variants and LoRA variants (Same experiment as above; including Table here too for better readability):
>
> | **Method** | **DomainNet** | **CORe50** |
> |------------|---------------|------------|
> | Generative Replay (Full FT) | 52.97 ± 0.07 | 86.28 ± 0.55 |
> | Generative Replay (LoRA) | 49.67 ± 0.21 | 83.64 ± 0.45 |
> | Generative Replay (LoRA - Matched) | 50.12 ± 0.09 | 84.21 ± 0.46 |
> | G2D (Full FT) | 54.37 ± 0.90 | 86.60 ± 0.28 |
> | G2D (LoRA) | **58.45 ± 0.56** | **89.11 ± 0.30** |
>
> This direct comparison highlights benefits of 0.32 ~ 8.66% improvement for the Full FT setting (see above Table and Tables 1,2,3,4) and 4.9 ~ 8.78% improvement for the LoRA setting (see above Table), which has *additional benefits of parameter efficiency*. We note that for *both* settings, our method still outperforms Generative Replay across all benchmarks.
>
> Further, our findings highlight how to better utilize recent parameter efficient fine-tuning techniques in continual learning, in that when learning *independent* weights, performance is often at least on-par with full fine-tuning. On the other hand, learning one set of parameters for multiple tasks can lead to performance drops due to this restriction of parameters; therefore, not being able to utilize PEFT solutions in an optimal fashion. Understanding this phenomena more in detail is an interesting direction for future work.

---

> ### Author Response · Authors · 2023-11-22
> **Response to Reviewer YtsH (3/3)**
>
> > For Generative Replay, I am surprised by the authors’ choice to “simply add the loss” of the replayed data to the loss on the current task. This creates an imbalance between G2D and Generative Replay. As described in subsection 3.2, the domain discriminator of G2D is trained on the union of synthetic data from all tasks so far...the domain discriminator is trained in a balanced way (i.e., all tasks so far are given equal weight). However, there is no such balance for Generative Replay. Because with Generative Replay, …, the loss of the classifier is constructed for 50% based on synthetic data from all past tasks and for 50% based on data from the current task…there is an imbalance (i.e., the last task is given higher weight than the previous tasks). I think it is likely this might negatively affect the performance of Generative Replay.
>
> We apologize for the confusion here. The classifier for Generative Replay is also *trained on the union of synthetic data from previous tasks and real data from current tasks (i.e., all tasks so far are given equal weight)*. This is what we meant by “in an unbiased manner” (i.e., correctly balancing the weights of all seen domains); otherwise, this would lead to a biased classifier. More explicitly, if we have seen t-1 tasks so far and currently training on t-th task, then that means we consider all t tasks with equal weight (1/t). Sorry again for the confusion, and we have stated this much more clearly in the revised draft (see Appendix C.3).
>
> Thank you again for your comprehensive and useful review! We hope the above clarifications and additional experiments addresses your concerns. Do you have any further questions? If you have any more suggestions or feedback, please let us know
>
> **References**
>
> [1] Classification accuracy score for conditional generative models. Ravuri et. al 2019.
>
> [2] CORe50: a New Dataset and Benchmark for Continuous Object Recognition. Lomonaco et. al 2017.

---

> > ### Comment · Reviewer_YtsH · 2023-11-23
> >
> > Thank you for the detailed response and clarifications. I'm afraid I do not have enough time to look at this in depth now, but I wanted to at least acknowledge your response before the discussion period ends.
> >
> > I can already acknowledge the comment about generative replay being also done on the union of synthetic data in a way that give equal weight to each task so far. Thank you for this clarification, and my apologies for mis-interpreting this in my previous comment.
> >
> > I will consider your other comments in depth later.

---

### Official Review · Reviewer_9jT2 · 2023-10-31

**Soundness:** 3 good
**Presentation:** 3 good
**Contribution:** 2 fair
**Rating:** 5
**Confidence:** 4

**Summary:**

The paper proposes tackles the setting of domain incremental learning by leveraging generative models as a routing mechanism. Specifically, for each task / domain $t$, the proposed approach
1. trains a domain specific expert $f_t$,
2. finetunes a pretrained generative model trained on $(x,y)$ pairs from task $t$,
3. trains a domain discriminator on the aggregated synthetic samples from all $t$ domains seen so far by sampling from the respective generative models.
4. At test time, the domain discriminator infers the task from the query data, and fetches the appropriate domain expert to make a prediction.

The authors evaluate the proposed method across four benchmarks, spanning both text and images, and real world medical imaging. Results show better performance than using the learned generative models for replay.

**Strengths:**

1. The approach is interesting; by decomposing the general domain incremental learning problem into (1) domain identification and (2) expert retrieval, the proposed approach is able to see performance gains.
2. The approach provides a fresh perspective on the use of synthetic data for domain incremental learning, which is potentially less vulnerable to sub-par generated samples.

**Weaknesses:**

1. The authors fail to discuss the computational cost of the method. How is the task discriminator trained ? Is it trained from scratch at every new domain, or continually learned ? What is the training cost of having to train two additional models (task classifier and generator) compared to expert learning ?
2. How does this approach scale ? My understanding is that it does so poorly if the task discriminator is not trained continually. More generally, it seems that the authors don't quite understand the computational efficiency related to PEFT approaches; taking LoRA for example, the computational cost saved from not performing a gradient update step on the full parameters is quite small compared to the cost of having to compute forward and backward passes in the model. The "gains" from peft are really in parameter efficiency and serving of these models.
3. Relevance of the setting : The authors provide initial motivation of the setting in the paper, where model weights may be made available, but not the actual data used for training. I have trouble seing healthcare institutions open-sourcing generative models of their data, but not the actual data itself. I would appreciate if the authors could point me to such instances.

**Questions:**

1. T5 is an encoder decoder model, thus enabling conditional generation. How are you generating synthetic data from this model, i.e. where is the data fed to the encoder coming from ? Do you have a separate generator for this ?
2. is the classifier at task t finetuned from task t -1 ? or finetuned from the pretrained model ?

---

> ### Author Response · Authors · 2023-11-16
> **Response to Reviewer 9jT2 (1/3)**
>
> Thank you for your thoughtful review! We will try to address your concerns below:
>
> > ...discuss the computational cost of the method. What is the training cost of having to train two additional models (task classifier and generator) compared to expert learning?
>
> Thank you for raising our attention to this. We have added the following paragraph to Appendix C.4, which details the computational cost of our method:
>
> “Our method incurs an additional total of 6-8 hours of computational cost on a single A6000 GPU, due to fine-tuning and sampling from the generator. Training the task discriminator, which takes less than 1-3 hours is done in parallel with training the expert classifier, so given that we permit the use of one more GPU, it does not incur additional compute time.
>
> This increase in compute cost (compared to expert learning), results in a substantial performance improvement - up to an absolute 8.1 point increase (see Table 4). In high-stakes applications such as healthcare, where missing even a single positive case (e.g., a fatal disease) could have critical consequences (e.g., a patient’s death), this performance gain is significant, and thus, we deemed this amount of tradeoff meaningful.”
>
>
>
> > How is the task discriminator trained? Is it trained from scratch at every new domain, or continually learned?
>
> The task discriminator is trained from the ViT-B16 pre-trained ImageNet checkpoint (vision) or BERT-base pre-trained checkpoint (text) at each domain.
>
> > How does this approach scale? My understanding is that it does so poorly if the task discriminator is not trained continually.
>
> As per your concern, we performed an additional experiment (added to Appendix B, Table 5 with a reference in the main text in section 3.2), where we train the task discriminator in a continual manner. Note that this turns the learning of domain discriminator into a class-incremental continual learning problem, introducing a new challenge on top of our original domain incremental learning problem.
>
> We find that the performance impact of continual training of the discriminator is minimal, implying that we *can continually train our discriminator model*. To assess the continuous training of the discriminator, we selected the dataset with the most number of domains, which is the CORe50 benchmark with a total of 8 different domains. This results in the performance of 88.13 ± 0.01 (averaged over 5 random seeds), while our original implementation results in 89.11 ± 0.30 (see Table 5). Thus. this continual fine-tuning of the domain discriminator results in a performance drop of less than 1 point. Therefore, our conclusions remain the same in that we *still* outperform existing state-of-the-art baselines by a substantial margin.
>
>
> > ... authors don't quite understand the computational efficiency related to PEFT approaches; taking LoRA for example, the computational cost saved from not performing a gradient update step … is quite small compared to … compute forward and backward passes …. The "gains" from peft are really in parameter efficiency …
>
> Our focus in elaborating on the LoRA fine-tuning techniques was to touch on parameter efficiency and not computational efficiency. While we have been careful to describe these methods as “parameter-efficient”, we realize there is a reference to “computational efficiency” in section 4.3 which can potentially confuse the reader. We apologize for this phrasing. To address this, we have updated the draft.

---

> ### Author Response · Authors · 2023-11-16
> **Response to Reviewer 9jT2 (2/3)**
>
> > … initial motivation of the setting in the paper, where model weights may be made available, but not the actual data used for training. I have trouble seing healthcare institutions open-sourcing generative models of their data, but not the actual data itself. … authors could point me to such instances.
>
> The setting where model weights may be shared but not the actual training data is a well-known setting in the healthcare domain [1][2][3]. We elaborate on two examples: (1) Kamran et. al 2021 presents a multisite external validation study for early identification of COVID-19 patients at risk of clinical deterioration, which requires sharing the model trained on private EHR data from one US hospital with 12 other US medical centers; (2) Ulloa-Cerna et. al 2022 presents a multisite external validation study for identifying patients at increased risk of undiagnosed structural heart disease, which requires sharing the model trained on private EHR data and patient echocardiography reports from one site with 10 other independent sites. While these are generally examples of discriminative models being shared across facilities as opposed to generative models, this demonstrates the general principle that in such domains, model sharing is often permissible in settings where data sharing is not.
>
> On one hand, as you note, it seems intuitive that healthcare institutions might be queasier about sharing generative models than sharing discriminative models.
>
> On the other hand,
> * Healthcare institutions are even queasier about sharing real data — and to this end, there is a **large mainstream line of work investigating the use of generative models for direct sharing** or for producing synthetic datasets that could be disseminated in lieu of actual patient data [4][5][6].
> * From the standpoint of most contractual or regulatory requirements, it is not yet clear if generative models sit in a different category than discriminative models or if they should just follow the same current regulatory requirements.
> * **How institutional practices develop and the regulatory environment evolve** will be informed, to a large degree, by exploratory research that characterizes both (i) the potential benefits and (ii) the potential risks associated with the dissemination of generative models trained on medical data. **We see our research as helping to elucidate the potential benefits**.
>
> We find this broad ongoing discussion on the utility of generative models in real-world deployment settings might be generally useful for the community. Therefore, we have updated the draft to include this ongoing discourse (see Appendix D).

---

> > ### Author Response · Authors · 2023-11-16
> > **Response to Reviewer 9jT2 (3/3)**
> >
> > > T5 is an encoder decoder model, thus enabling conditional generation. How are you generating synthetic data from this model, i.e. where is the data fed to the encoder coming from ? Do you have a separate generator for this?
> >
> > In Appendix C.2, we have listed implementation details about the T5 generator model. For our generator, we use the prompt tuning (Lester et al., 2021) to learn the parameter-efficient models. We use the pre-trained T5-Large v1.1 checkpoint adapted for prompt tuning as the backbone (Raffel et al., 2020), and the prompt embeddings are initialized randomly. We set the prompt length to 400 tokens which accounts for 819K trainable parameters, i.e., around 0.1% in comparison to 784M frozen T5-Large parameters. During training, we input a special token into the model and conditionally generate a document content, question, and answer, all separated by the special tokens. Our maximum sequence length is set to 512, and we truncate the document content after tokenizing the question-answer pair. During the generation process, we provide multiple text prompts along with the special input token and learned prompt embeddings. We do not use any generator to sample our input text prompts and consider the following text prompts to conditional generate synthetic samples – “Generate article, question and answer.”, “Generate context, question and answer.”, “Generate answers by copying from the generated article.”, “Generate factual questions from the generated article.”. During generation, we use ancestral sampling, which selects the next token randomly based on the model’s probability distribution over the entire vocabulary, thereby reducing the risk of repetition. We generate samples with a minimum length of 50 tokens and a maximum of 1,000 tokens, retaining only those samples that contain exactly one question-answer pair with the answer included in the generated document content.
> >
> > > is the classifier at task t finetuned from task t -1 ? or finetuned from the pretrained model ?
> >
> > The classifier is continually fine-tuned from task t-1.
> >
> >
> > We hope these answers and additional experiments help clarify your concerns. We would be happy to answer any further questions you have. If you do not have any further questions, we hope that you may consider raising your score. Thank you again for your constructive feedback!
> >
> >
> > **References**
> >
> > [1] Early identification of patients admitted to hospital for covid-19 at risk of clinical deterioration: model development and multisite external validation study. Kamran et. al 2021.
> >
> > [2] rECHOmmend: An ECG-Based Machine Learning Approach for Identifying Patients at Increased Risk of Undiagnosed Structural Heart Disease Detectable by Echocardiography. Ulloa-Cerna et. al 2022.
> >
> > [3] Development and Multi-Site External Validation of a Generalizable Risk Prediction Model for Bipolar Disorder. Waish et.al 2023.
> >
> > [4] Synthetic data in machine learning for medicine and healthcare. Chen et. al 2022.
> >
> > [5] Synthetic Medical Images for Robust, Privacy-Preserving Training of Artificial Intelligence. Coyner et. al 2022.
> >
> > [6] Overcoming barriers to data sharing with medical image generation: a comprehensive evaluation. Schutte et. al 2021.

---

> > > ### Comment · Reviewer_9jT2 · 2023-11-16
> > > **Discussion**
> > >
> > > Thank you for the detailed answers. The authors have provided reasonable answers to my initial concerns about the paper. While I do have some concerns about how the potential use cases of such an approach, I acknowledge the author's answers and will update my score as such.

---

> > > > ### Author Response · Authors · 2023-11-20
> > > > **Reply to Reviewer 9jT2**
> > > >
> > > > Thank you for your time and continued engagement! We are happy to have addressed your initial concerns. We also understand that there are potential challenges in medical use cases, based on how institutional healthcare practices develop and the regulatory environment evolves. Could you elaborate more on your remaining concerns in this area so that we can properly address or acknowledge those as well before the rebuttal period ends? Thank you again for your constructive feedback!

---

### Author Response · Authors · 2023-11-16
**Overall response**

We thank all the reviewers for their valuable time. We are glad to see that all reviewers agreed that the idea offers a “fresh and neat perspective” on the use of synthetic data in continual learning (9jT2, YtsH, 833z). Furthermore, we are glad to see reviewers have highlighted our approach as “interesting” (9jT2), “an important, insightful contribution” (YtsH), and the “extensiveness of experimentation with various datasets and multiple modalities (image, text)” (833z).

We have added additional experiments, clarifications, and minor revisions to the paper which are highlighted in blue in the text.

We respond to each reviewer’s comments individually, including additional experiments we have conducted to enhance the strength of our paper.

---

### Meta-Review · Area_Chair_Uoy2 · 2023-12-10

**Metareview:**

The paper proposes an approach to tackle the problem of catastrophic forgetting across multiple different domains. The main idea is to train a domain-conditional generative model and a domain conditional prediction function, then use the samples from the generative models to train a domain discriminator. At test time, the domain discriminator is used to select the domain conditional prediction function to use for the query. On four benchmarks, the main finding is that using synthetic data to train a domain discriminator followed by a decision function works better than using the same synthetic samples to augment ERM. Overall the reviewers found this paper a very interesting idea and liked the direction it took. However at the end of the review period, there was still uncertainty about the manuscript as it stands. In particular, reviewers found that the initial submission lacked several important experimental details -- while the reviewer discussion period did clarify some of them, there still remained concerns based on the asymmetric use of LoRA. To address this the authors ran additional experiments during the rebuttal period where they studied what happens when learning G2D with full fine-tuning and running Generative Replay with LORA.They found that G2D improved performance w/ LORA but Generative Replay dropped in performance -- not all reviewers were satisfied with this explanation. The authors proposed that this arose due to the inherently more complex problem that Generative Replay was tackling -- i.e. full fine tuning works better because the model has to learn to solve all tasks simultaneously but it worked less well in G2D (and LORA works better) because full fine-tuning could be susceptible to overfitting. This seems a sensible explanation for the phenomena but the reviewers wanted further empirical validation that this explanation was what was driving the empirical results.

For my own part, I thought the paper does indeed present an interesting finding but given that much of this work's push is driven by empiricism, I think additional experimentation to cement the rationale behind the design choices is warranted.

While the reviewers did not bring this up, I encourage the authors to contemplate how their work differs from other research that attempts to learn routing functions (e.g. Routing networks and their variants: see https://github.com/cle-ros/RoutingNetworks?tab=readme-ov-file). Comparing the G2D approach against a variant of Generative Replay based on Routing Networks  would significantly strengthen their claim since it would assess the degree of improvement in a test time routing procedure relative to a learned one at training time since some portion of the reduction in scores from Generative Replay could be based off of not being able to correctly learn a routine mechanism at training time (the precise problem considered in the suggested paper https://openreview.net/forum?id=ry8dvM-R-).

**Justification For Why Not Higher Score:**

The need for further ablation results pointed out by the reviewers.

**Justification For Why Not Lower Score:**

N/A

---

### Decision · Program_Chairs · 2024-01-16

Reject